# Optical deciphering of multinary chiral compound mixtures through organic reaction based chemometric chirality sensing

Diandra S. Hassan[1] & Christian Wolf [1✉]

The advances of high-throughput experimentation technology and chemometrics have revolutionized the pace of scientific progress and enabled previously inconceivable discoveries, in particular when used in tandem. Here we show that the combination of chirality sensing based on small-molecule optical probes that bind to amines and amino alcohols via dynamic covalent or click chemistries and powerful chemometric tools that achieve orthogonal data fusion and spectral deconvolution yields a streamlined multi-modal sensing protocol that allows analysis of the absolute configuration, enantiomeric composition and concentration of structurally analogous—and therefore particularly challenging—chiral target compounds without laborious and time-consuming physical separation. The practicality, high accuracy, and speed of this approach are demonstrated with complicated quaternary and octonary mixtures of varying chemical and chiral compositions. The advantages over chiral chromatography and other classical methods include operational simplicity, increased speed, reduced waste production, low cost, and compatibility with multiwell plate technology if high-throughput analysis of hundreds of samples is desired.

[1] Department of Chemistry, Georgetown University, Washington, DC 20057, USA. ✉email: cw27@georgetown.edu

Chiral compounds are ubiquitous in nature and play essential roles in the chemical, environmental, materials, and health sciences. Paradigm-shifting advances with an asymmetric synthesis of chiral compounds have become possible with the aid of artificial intelligence but there has been a little impact at the forefront of enantioselective analysis[1–6]. Non-racemic mixtures containing both enantiomers sometimes in vastly varying ratios are frequently encountered in natural samples or during chemical development and need to be analyzed both accurately and quickly. The determination of the total amount and enantiomeric composition of chiral compounds has therefore become a recurring but often challenging task in literally countless R&D projects. This can entail a prohibitively laborious and time-consuming process, in particular when the sample is a mixture of several compounds—a fairly common scenario. To circumvent the daunting complexity of enantioselective multicomponent analysis, each chiral analyte is typically first isolated to determine its amount and the enantiomeric ratio (er) is then uncovered in a separate experiment by chromatography on a chiral stationary phase or by NMR spectroscopy with a chiral derivatizing or solvating agent. The limitations of classical enantioselective analysis become increasingly apparent when many samples consisting of several chiral analytes in varying concentrations and enantiomeric compositions need to be examined. To date, chiral chromatography, which is intrinsically serial because one can only run one sample at a time, remains the workhorse and is widely considered the gold standard. Despite the development of fast and two-dimensional chromatographic methods, a general solution amenable to high-throughput experimentation and parallel chiral multicomponent analysis is not in sight.

During the last decade, many chiroptical sensing methods that are compatible with automation, multiwell plate technology, and parallel data acquisition have been introduced to address the shortcomings of classical techniques and to improve throughput, time-efficiency, and sensitivity at reduced cost, waste production, and energy consumption[7–9]. Molecular and supramolecular sensor arrays mimicking chemical nose detection have been used for qualitative analysis of chiral compound mixtures[10–12]. The potential of quantitative enantioselective optical sensing is exemplified by several case studies in which both the er and the concentration of a single analyte were determined with carefully designed UV, fluorescence, and circular dichroism (CD) probes[13–19]. Alternatively, single compound er determination has been accomplished with linear discriminant analysis, artificial neural network, and principal component analysis[20–23]. Recently, chiroptical er/dr sensing methods have been reported and applied to quantitative analysis of stereoisomeric mixtures of amino alcohols with two chiral centers.[24,25] We hypothesized that merging state-of-the-art chiroptical sensing methodology and artificial intelligence for spectral deconvolution would overcome long-standing difficulties and limitations of traditional chromatography based approaches and provide a solution toward comprehensive (determination of absolute configuration, enantiomeric ratio, and total concentration) in situ analysis of complicated multicomponent mixtures without physical separation. To this end, major obstacles originate from the difficulty with quantitative deconvolution of a massive amount of spectroscopic data that would be generated by simultaneous sensing of several chiral analytes and the low resolution of inherently broad and largely overlapping CD and UV absorption bands. We envisioned that this can be addressed by integrating robust, broadly applicable chiroptical sensing technology and chemometric tools capable of deciphering highly convoluted, multi-modal spectral information.

Herein we show that simultaneous analysis of individual concentrations and er's of complicated multinary chiral compound mixtures is possible by using UV and circular dichroism data obtained by organic reaction-based optical sensing with a single achiral probe. The demonstrated practicality and speed of chemometric analysis-based data fusion and spectral deconvolution of quaternary and octonary samples are expected to largely alter how chiral compound development and analysis tasks are solved and bear the potential to streamline the workflow in numerous academic and industrial laboratories.

## Results and discussion

**Organic reaction-based chirality sensing.** At the onset of this study, we selected 1-phenylethylamine (**PEA**), 1-(pyrrolidin-2-ylmethyl)pyrrolidine (**PMP**), phenylglycinol (**PGL**) and 2-amino-1-phenylpropan-1-ol (**PPA**) to structurally represent frequently encountered chiral amine and amino alcohol drugs, auxiliaries, and synthetic building blocks (Supplementary Fig. 1). Another important selection criterion was to assemble a challenging mixture of similar compounds exhibiting the same functionalities and identical or closely related carbon scaffolds that either contains small aromatic moieties or are purely aliphatic (Fig. 1a). These chiral analytes are difficult to detect and quantify, even in enantiopure form. In fact, we observed negligible UV and CD signals at low wavelengths when 0.083–0.125 mM solutions in 1,2-dichlorethane (DCE) were analyzed (Supplementary Figs. 2 and 3). To enable chiroptical sensing we, therefore, screened five probes that operate on the principles of irreversible covalent chemistry (ICC) or dynamic covalent chemistry (DCC). The 4-chloro-3-nitrocoumarin (**A**) and N-(5-fluoro-2,4-dinitrophenyl)benzamide (**B**) belong to the first group while 2,4-dinitrobenzaldehyde (**C**), salicylaldehyde (**D**), and ninhydrin (**E**) undergo reversible Schiff base or acetal formation (Supplementary Figs. 4 and 5)[26–29]. Regardless of their operational mode, all five probes react quickly with the target compounds at room temperature and thus introduce a strong chromophore close to the chiral center (Fig. 1b). This allows optical visualization through an operationally simple mixing protocol that does not require any precautions as exposure to air and moisture does not interfere with the sensing chemistry. Another noteworthy feature is that these probes do not generate a new chiral center which avoids increasing molecular and analytical complexity that would arise from the formation of diastereomeric species. The suitability of these probes for the daunting task of combined concentration and er analysis with multinary compound mixtures was first assessed individually with (S)-**PEA**, (S)-**PMP**, (S)-**PGL,** and (1R,2S)-**PPA**, respectively (Fig. 1b and Supplementary Figs. 6–17). We found that the coumarin probe **A** generates distinct CD signals with all four analytes and particularly strong Cotton effects with the amines **PEA** and **PMP**. The substrate binding to **A** also produces characteristic UV changes that in all cases are quite different from the original UV signature of this sensor. The optical sensing with the other probes showed less promise due to striking similarities and in some cases considerable signal overlapping across the whole spectral range from 250 to 450 nm. For example, optical visualization of **PEA**, **PGL,** and **PPA** with **B** gave essentially identical UV spectra. The CD effects were also unsatisfactory. While we measured strong CD signals upon binding of (S)-**PMP**, the signal induction for (1R,2S)-**PPA** was very weak, and (S)-**PEA** and (R)-**PGL**, which only differ by the presence of the terminal alcohol group but otherwise share the same 3D carbon scaffold, gave superimposable spectra (please note that (S)-**PEA** and (R)-**PGL** have identical three-dimensional structures but different stereochemical descriptors only by virtue of the CIP rules). Similar problems, i.e., considerable spectral overlap across the 250–450 nm region, were encountered with probes **C**, **D**, and **E** which underscores the difficulty of the molecular and chirality recognition tasks with the selected amines and amino alcohols. In fact, the aromatic aldehyde sensors **C** and **D** are only applicable

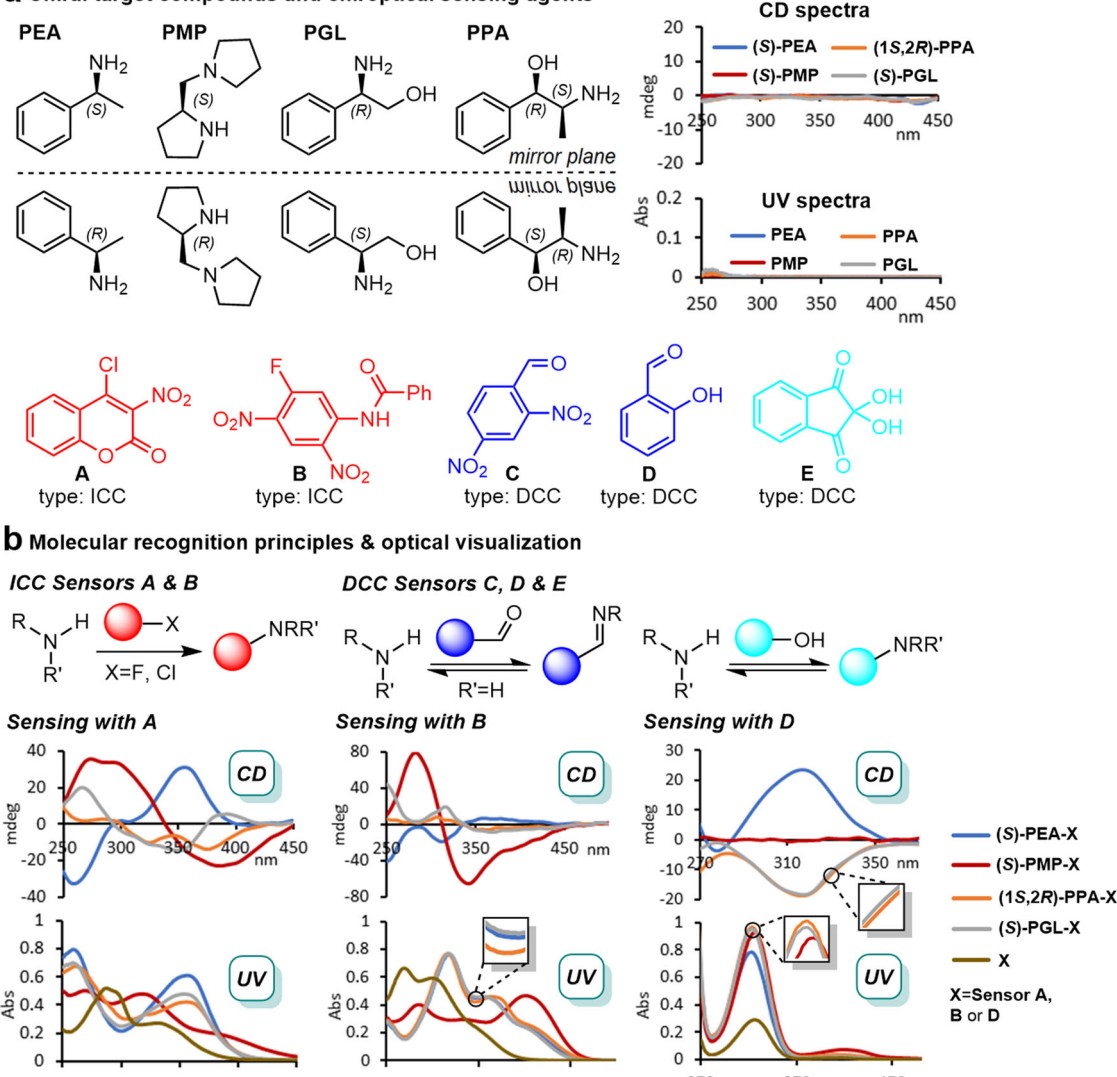

**Fig. 1 Organic reaction-based chirality sensing. a** Structures of the target compounds with inherently negligible optical activity. CD and UV spectra of (*S*)-**PEA**, (*S*)-**PMP**, (*S*)-**PGL** and (1*R*,2*S*)-**PPA** were recorded in 1,2-dichloroethane (DCE) at 0.125 and 0.083 mM, respectively. Chiroptical sensing agents operating based on irreversible covalent chemistry (ICC type) and dynamic covalent chemistry (DCC). **b** Molecular recognition principles and induced UV and CD spectra. Sensing with **A**: UV 0.094 mM, CD at 0.042 mM in DCE; **B**: UV 0.083 mM, CD at 0.021 mM in CHCl₃; **D**: UV 0.062 mM, CD at 0.125 mM in DMSO. **PEA**: 1-phenylethylamine, **PMP**: 1-(pyrrolidin-2-ylmethyl)pyrrolidine, **PGL**: phenylglycinol, **PPA**: 2-amino-1-phenylpropan-1-ol.

to analytes with a primary amino group and fail to generate a CD signal with the enantiomers of **PMP**, which cannot form a Schiff base.

**Quantitative chirality sensing based on complementary CD induction.** Altogether, this clearly pointed towards coumarin **A** as the optimal chiroptical sensor choice. In order to validate the utility of this probe, we applied it to each enantiomer under identical conditions (Fig. 2a). As expected, we obtained opposite induced CD (ICD) signals for the enantiomeric pairs which we anticipated can be used for the determination of the enantiomeric ratio (*er*). The individual UV changes described above were predicted to correlate with the total concentration of each target compound because this optical response is essentially the same for either enantiomer and independent of the *er*. Our first inspection of the CD responses generated by sensing with **A** revealed that the spectra produced with the four analytes would largely overlap with the exception of **PEA** and **PMP**, at least at

two wavelengths (Fig. 2b and Supplementary Figs. 18–21). The ICDs of the enantiomeric **PMP-A** adducts display *x* axis intercepts at 340 nm, a wavelength at which the **PEA-A** enantiomers show relatively strong CD responses (Supplementary Fig. 22). By contrast, the enantiomeric **PEA-A** adducts are CD-silent at 400 nm which is close to a local ICD maximum produced by coumarin sensing of (*R*)- or (*S*)-**PMP**. We envisioned that this should present a rare opportunity for concomitant *er* sensing of these two amines in a single sample unless there is interference with the derivatization step or the chiroptical amplification[30]. We prepared ten mixtures containing equimolar amounts of **PEA** and **PMP** in largely varying enantiomeric compositions for sensing with **A** (Supplementary Figs. 23–32 and Supplementary Table 1). The CD responses of this probe at 340 and 400 nm in the corresponding spectra were then used for linear regression analysis. We found perfectly linear correlations between the ICD amplitudes generated by **A** and the enantiomeric compositions of the two amines which allowed accurate *er* analysis while the absolute

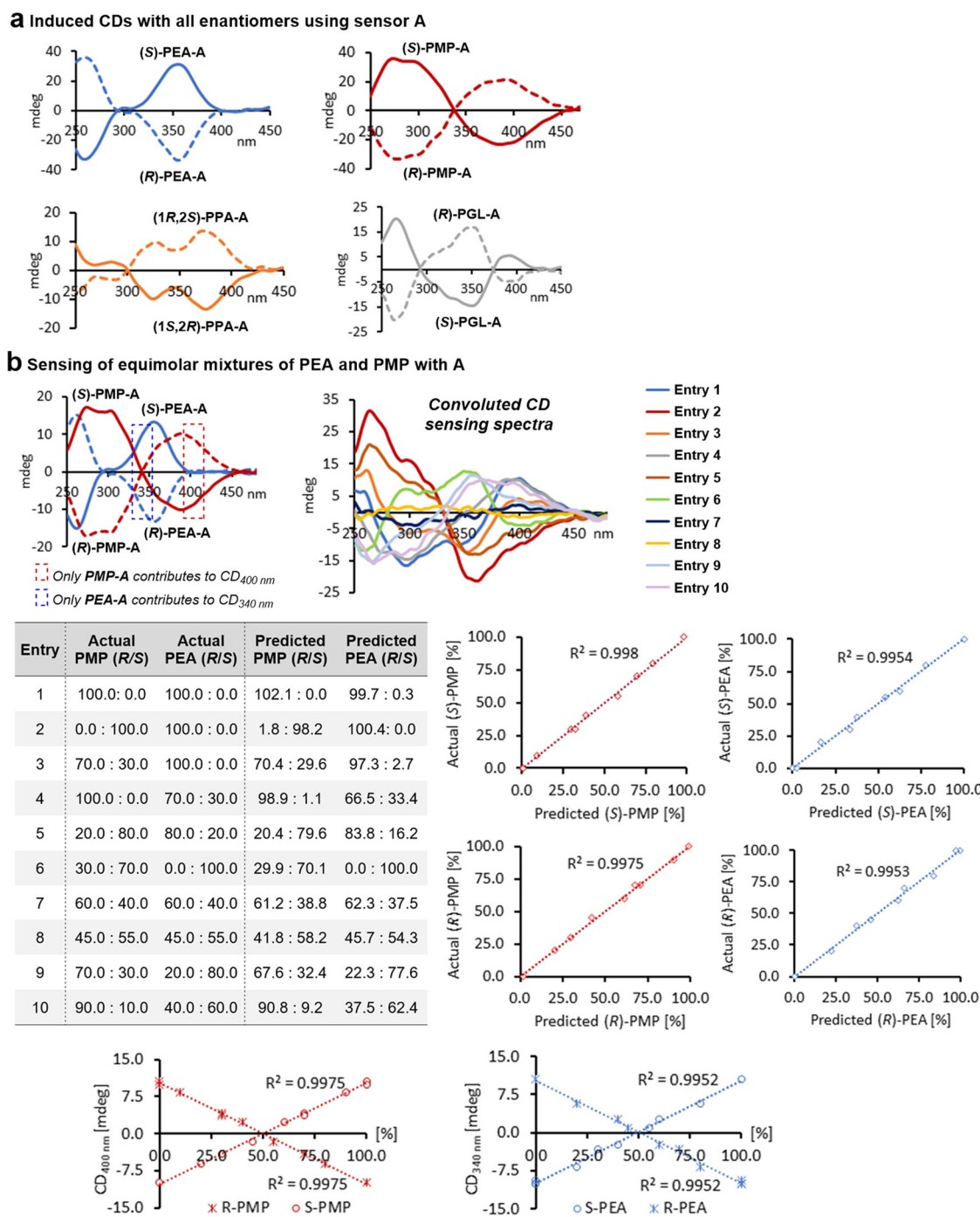

**Fig. 2 Quantitative chirality sensing based on complementary CD induction. a** ICD sensing of all enantiomeric pairs using **A**, all measurements were taken at 0.094 mM in DCE. **b** Sensing of equimolar mixtures containing nonracemic **PEA** and **PMP** with **A**. The reactions were performed at 12.5 mM in DCE and CD measurements were taken at 0.075 mM in the same solvent. **PEA**: 1-phenylethylamine, **PMP**: 1-(pyrrolidin-2-ylmethyl)pyrrolidine.

configuration of the major enantiomer was determined by comparison of the (±)-sign of the ICD with a reference sample (Supplementary Figs. 33 and 34). For example, the *er*'s of a sample containing the (*R*)- and (*S*)-enantiomers of **PMP** and **PEA** in 70.0:30.0 and 100.0:0.0 ratios, respectively, were determined as 70.4:29.6 and 97.3:2.7 (Fig. 2b, Table entry 3). In another case, the sensing of a sample composed of 20.0:80.0 and 80.0:20.0 of the (*R*)- and (*S*)-enantiomers of **PMP** and **PEA** gave 20.4:79.6 and 83.8:16.2 (entry 5). All chiroptical sensing results are within a relatively small absolute error margin of <4% which is generally acceptable, in particular in high-throughput screening applications where error margins of 5–10% have been considered satisfactory (Supplementary Figs. 35–38)[9,31]. These initial studies demonstrated to us the suitability of the coumarin probe **A** for sensing of the absolute configuration and *er* of mixtures of **PEA** and **PMP**. This chiroptical assay is very practical, fast (it is complete within 15 min), yields strong ICD effects that increase linearly with the analyte *er*, and does not show any chemical interferences. A scenario in which complementary CD responses at carefully selected wavelengths are generated during sensing of two analytes as depicted above, however, is rare and the traditional chiroptical data handling applied is not suitable for samples

of which the enantiomeric compositions and analyte concentrations are unknown because both affect the observable CD induction. We therefore resorted to investigating chemometric tools for such a task[32].

**Chemometric concentration and _er_ determination with quaternary mixtures.** Traditional univariate CD sensing of chiral compounds allows quantification of the enantiomeric sample composition if the total concentration is known or experimentally determined with a separate technique. This can, for example, be achieved with UV spectroscopic analysis as is often the case when chiroptical methods are used because modern circular dichroism spectrophotometers generate CD and UV spectra in parallel. When the analyte concentration, i.e., the total amount of both enantiomers, is known the measured CD amplitude at a given wavelength—preferentially where the CD spectrum has a maximum—can be directly correlated to the enantiomeric ratio (_er_) with the help of a calibration curve. In special cases, complementary pairs of induced CD spectra are obtained during sensing of mixtures of two chiral compounds as shown above. If the total concentrations of the chiral targets are known one can use the CD values at carefully selected wavelengths where only one of the analytes contributes to the chiroptical sensor response to quantify the enantiomeric ratios of each compound step-by-step. A more typical analytical scenario, however, is that compound concentrations and enantiomeric ratios vary and need to be determined. To this end, it is important to take into consideration that sensing of two samples with different concentrations (or enantiomeric ratio) but the same enantiomeric ratio (or concentration) is likely to yield completely different CD outputs. In addition, spectra of compound mixtures are often highly convoluted which precludes univariate analysis. Because both the _er_ and the concentration affect the CD signals induced by the sensor the chiroptical analysis becomes a complicated multivariate problem with substantially overlapping spectra that cannot be solved by traditional approaches (Fig. 3 and Supplementary Fig. 51). Univariate optical sensing also suffers from other drawbacks that limit its utility. The restriction to single-wavelength analysis discards most information contained in the spectrum. For example, if an induced CD spectrum consists of 250 data points, using only a single wavelength for linear regression analysis exploits just 0.4% of the whole spectral information while the remaining 99.6% are literally ignored, which makes it more susceptible to chemical and optical interferences.

Since the absolute configuration, enantiomeric ratio, and concentration of chiral compound mixtures that generate highly convoluted chiroptical spectra cannot be comprehensively determined by single-wavelength analysis, we resorted to chemometric tools which have become increasingly popular for processing of large data sets that overwhelm traditional data handling approaches[33]. In multivariate analysis, multiple independent variables are considered to minimize information loss and possible interference from spectral noise. We, therefore, decided to apply multivariate chemometric sensing to quaternary mixtures containing the enantiomers of **PEA** and **PMP** in varying concentrations and enantiomeric ratios (Fig. 3 and Supplementary Figs. 39–48, 52–93). In full spectrum analysis, the number of independent variables (the number of CD and UV data points included, e.g., the whole range from 250 to 450 nm) is typically larger than the number of dependent variables (how many samples are in a training set), and simple ordinary least square methods without prior dimensionality reduction cannot be used. A useful approach to full-spectrum analysis is Latent Variable Multivariate Regression (LVMR), where the dimensionality of the

data is reduced and regression is then applied to model the relationship between the dependent and independent variables. When data are reduced to a lower dimension, one concern is information loss. To examine this with quaternary chiral compound mixtures, we applied principal component analysis (PCA) to the convoluted UV and CD spectra obtained with our coumarin sensing assay. PCA is an unsupervised dimensionality reduction technique that projects large data sets into smaller ones, called principal component (PC), in a way that maximizes the variance, i.e., the overall variability of the data. Figure 3 contains an illustration of how PCA can transform a 2D plot with correlated data points into one dimension. PCA is very useful for large, collinear data sets, like those obtained in UV and CD spectroscopy, where the independent variables are continuous and therefore highly correlated (see Supplementary Table 2 for explained variance ratio for CD and UV data). Using PCA, we were able to reduce the convoluted CD and UV spectra obtained by our chiroptical sensing assay to four PCs and then reconstructed the spectra via inverse transformation (Fig. 3 and Supplementary Figs. 49, 50). Plotting of the original and the reconstructed spectra revealed almost perfect overlap, demonstrating that there is minimal information loss after reducing the dimensionality of the original spectral data. In other words, most of the information contained across the whole spectrum range is conserved in the PC.

One simple yet powerful LVMR method is Principal Component Regression (PCR), which is a multivariate calibration technique that combines PCA with Ordinary Least Squares (OLS) principles. We constructed a regression model using PCR with 16 training sets and 5 test sets (Fig. 4). First, PCA treatment of the UV and CD spectra allowed us to reduce the dimensionality into 4 PCs, and then OLS was applied to model the analyte concentrations using the PCs as independent variables. Cross-validation was performed on the training sets to evaluate how well the model can be applied to new data. The heat map in Fig. 4 displays how accurately the algorithm predicts the quaternary sample compositions (Supplementary Figs. 94 and 95). The actual millimolar concentrations of the enantiomers of **PEA** and **PMP** are given in each box while the colors correspond to the sensing accuracy. The green color indicates a relatively small absolute deviation of <0.5 mM that increases up to 1.5 mM represented by yellow coloring. The PCR algorithm used shows excellent overall performance (averaged $R^2 = 0.94$ and RMSE = 0.52 during cross-validation; averaged $R^2 = 0.99$ and RMSE = 0.17 for the test sets, see Supplementary Figs. 96 and 97), and the successful correlation between all actual and predicted concentration values as depicted in the scatterplot in Fig. 4 underscores the general utility. We like to point out that with a trained model in hand the PCR test sample analysis, programmed in Python or another programming language (R, Matlab, etc), is completed within seconds using a simple desktop computer. A detailed comparison between the actual and the predicted individual concentrations of the four chiral compounds in the five test sets is visualized in the stacked column chart which shows that the results obtained with our chemometric coumarin sensing analysis are very close to the original sample compositions. Altogether, this demonstrates reliable determination of absolute configuration, _er_ and concentration of complicated quaternary mixtures by a straightforward chemometric sensing protocol. For example, the concentrations of the **PEA** and **PMP** enantiomers in the test sample #4 were 3.75 mM for (_R_)-**PEA**, 1.25 mM for (_S_)-**PEA**, 11.25 mM for (_R_)-**PMP**, and 3.75 mM for (_S_)-**PMP**, which compares well with the chemometrically predicted concentrations of 3.82, 1.32, 11.35, and 3.51 mM, respectively. As expected, the error of chemometric chirality sensing of samples with low enantiomeric ratios or even racemates is increased due to weaker induced CD signals with a

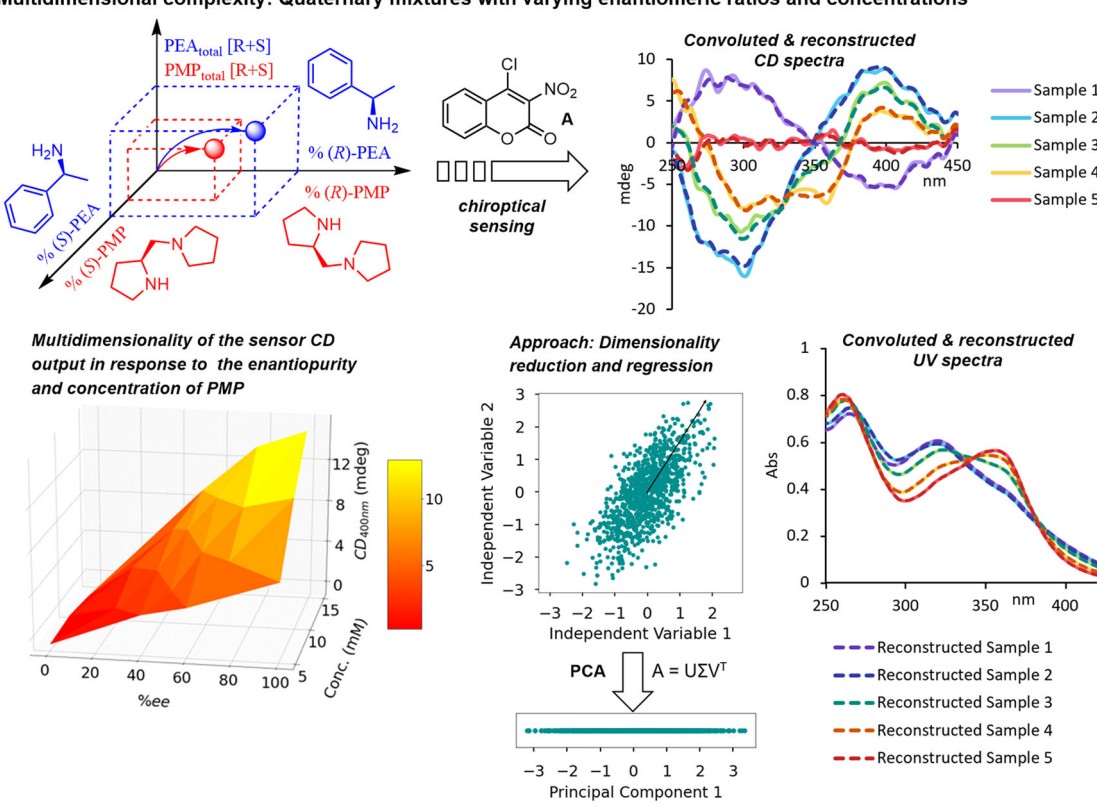

**Fig. 3 Multidimensional complexity of quaternary mixtures of the individual enantiomers of PEA and PMP in varying concentration and enantiomeric composition.** The dependence of the induced CD signal on the sample concentration and enantiopurity is shown exemplarily for PMP in a 3D plot. The convoluted UV and CD spectra of five samples and the reconstituted spectra were subjected to PCA (principal component analysis); A is the m × n matrix. U is the Eigenvector of $A^T A$. $V^T$ is the Eigenvector of $AA^T$. Σ contains the square roots of the Eigenvalues. **PEA**: 1-phenylethylamine, **PMP**: 1-(pyrrolidin-2-ylmethyl)pyrrolidine. The data underlying this figure are available as Source Data.

higher signal-to-noise ratio. This is exemplified by the training set #13 containing equimolar amounts of both enantiomers of **PEA** and **PMP** at 5.50 and 4.50 mM, respectively. The absolute errors were determined as 1.65 mM for (R)-**PEA**, 1.13 mM for (S)-**PEA**, 0.51 mM for (R)-**PMP**, and 1.04 mM for (S)-**PMP** as indicated by the light green to yellow colors in the heat map. This contrasts with the training set #16 containing enantiopure (R)-**PEA** and (R)-**PMP** at 7.00 and 13.00 mM, respectively, and thus producing relatively strong ICD signals upon reaction with the coumarin probe. In this case, the absolute errors are only 0.08 mM for (R)-**PEA**, 0.29 mM for (S)-**PEA**, 0.28 mM for (R)-**PMP** and 0.00 mM for (S)-**PMP**.

**Multi-modal optical chirality sensing of octonary samples.** Finally, we decided to attempt chemometric sensing of octonary chiral compound mixtures by exploiting multi-modal analysis tools that are capable of processing multivariate data sets from complementary measurements. As explained above, we anticipated highly convoluted spectra because of the structural similarity of **PEA**, **PMP**, **PGL**, and **PPA**, which were intentionally chosen to allow evaluation of the capabilities, limitations, and robustness of chemometric organic reaction-based chiroptical sensing. We realized early on that this increasingly complicated task could not be adequately performed by traditional methods and would instead require adaptation of multiblock chemometrics which, for example, have been successfully used to integrate Raman and IR spectral data sets for comprehensive materials characterization[34,35]. To obtain multi-modal data we favored a practical solution that is based on essentially the same

chiroptical sensing procedure with coumarin **A** described above but in different solvents. The CD and UV sensing spectra of octonary mixtures of the enantiomers of **PEA**, **PPA**, **PGL**, and **PMP** were collected in dichloroethane and methanol, respectively, with the expectation that these solvent choices would induce distinguishable chiroptical coumarin responses due to variance in intramolecular hydrogen bonding and altered conformational equilibria (Fig. 5a and Supplementary Figs. 98–199). Our initial optical sensing analysis proved very promising and showed that this can indeed increase spectral information. For example, we observed very similar UV spectra by coumarin sensing of **PEA** and **PPA** in MeOH which turned strikingly different when dichloroethane was used as a solvent. As expected, the ICD spectra obtained with **PMP** in these two solvents are very similar, presumably because **PMP**-**A** does not exhibit an intramolecular hydrogen bond which largely reduces the solvent dependence of the CD readout. By contrast, the coumarin derivatives of the other compounds, in particular, **PEA**-**A** and **PPA**-**A**, display significant ICD modifications as the solvent is changed (Fig. 5a and Supplementary Figs. 6–9).

First, octonary mixtures were subjected to coumarin sensing and traditional PCA and PLS modeling. Reconstruction of the UV and CD spectra showed minimal data loss (Fig. 5b). We then used two multiblock algorithms, MBPCA and MBPLS, for regression analysis of the octonary mixtures. MBPCA and MBPLS are the multiblock versions of PCA and PLS, respectively, and both algorithms aim to increase the interpretability of multivariate models. For both cases, the interpretability is exemplified by the block loadings that inform how much individual wavelengths contribute to the latent variable/principal

**Quantifying the concentration and enantiomeric ratio of complex quaternary mixtures**

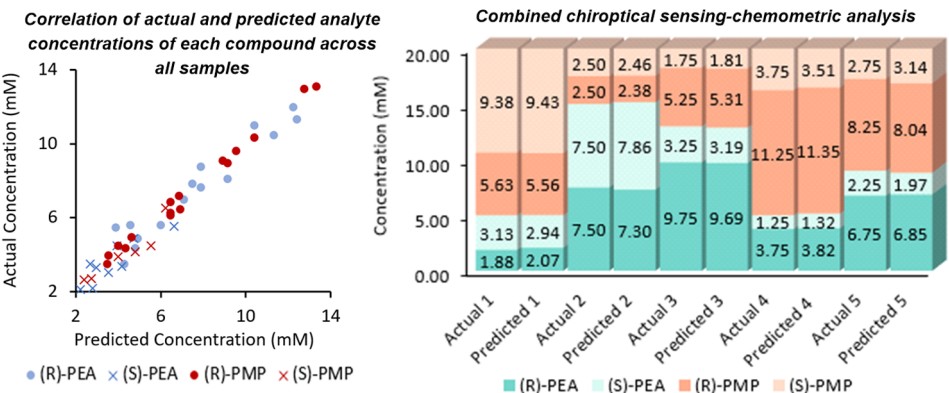

**Heat map showing the sensing accuracy**

| Sample | (R)-PEA (mM) | (S)-PEA (mM) | (R)-PMP (mM) | (S)-PMP (mM) |
|---|---|---|---|---|
| Training Set 1 | 5.60 | 1.40 | 10.40 | 2.60 |
| Training Set 2 | 7.70 | 3.30 | 6.30 | 2.70 |
| Training Set 3 | 3.50 | 3.50 | 6.50 | 6.50 |
| Training Set 4 | 10.50 | 4.50 | 3.50 | 1.50 |
| Training Set 5 | 11.38 | 1.63 | 6.13 | 0.88 |
| Training Set 6 | 11.00 | 0.00 | 9.00 | 0.00 |
| Training Set 7 | 4.38 | 0.63 | 13.13 | 1.88 |
| Training Set 8 | 7.88 | 1.13 | 9.63 | 1.38 |
| Training Set 9 | 15.00 | 0.00 | 5.00 | 0.00 |
| Training Set 10 | 4.90 | 2.10 | 9.10 | 3.90 |
| Training Set 11 | 8.80 | 2.20 | 7.20 | 1.80 |
| Training Set 12 | 5.63 | 3.38 | 6.88 | 4.13 |
| Training Set 13 | 5.50 | 5.50 | 4.50 | 4.50 |
| Training Set 14 | 8.13 | 4.88 | 4.38 | 2.63 |
| Training Set 15 | 12.00 | 3.00 | 4.00 | 1.00 |
| Training Set 16 | 7.00 | 0.00 | 13.00 | 0.00 |
| Sample 1 | 1.88 | 3.13 | 5.63 | 9.38 |
| Sample 2 | 7.50 | 7.50 | 2.50 | 2.50 |
| Sample 3 | 9.75 | 3.25 | 5.25 | 1.75 |
| Sample 4 | 3.75 | 1.25 | 11.25 | 3.75 |
| Sample 5 | 6.75 | 2.25 | 8.25 | 2.75 |

Numbers show the sample concentrations. The colors represent the absolute deviation of the ML analysis.

**Fig. 4 Chiroptical sensing of quaternary mixtures.** Heat map for the 16 training sets and the 5 samples. Color corresponds to the absolute error of predicted [mM] versus actual [mM]. The sensing-chemometrics analysis results are compared to the actual data in the stacked column chart. **PEA**: 1-phenylethylamine, **PMP**: 1-(pyrrolidin-2-ylmethyl)pyrrolidine. The data underlying this figure are available as Source Data.

components LV1-4 or PC1-4 (Fig. 5b and Supplementary Tables 14, 15, 20–45, Supplementary Figs. 208–211). Interestingly, both algorithms select similar wavelength intervals with the strongest contributions for the first LV and PC, respectively, i.e., 270–320 nm and 360–430 nm for CD; 270–330 nm and 400–450 nm for UV. Evaluation of the block importance in MBPLS analysis of (R)-**PEA** revealed that the CD data contribute 64% to LV1, a general trend also observed with the other compounds (see pie charts in Fig. 5b and Supplementary Figs. 208–211). For the first and second latent variable, the strongest contributions come from CD data in MeOH and DCE while the third and fourth latent variable largely depends on UV data obtained in DCE. In order to maximize accuracy and minimize overfitting at the same time, the optimal number of latent variables/principal components having the smallest RMSE was determined by leave-one-out cross-validation (LOOCV) (Supplementary Figs. 200–207). For example, three and four LVs were used for the analysis of (S)-**PMP** and (R)-**PEA**, respectively (Fig. 5b and Supplementary Figs. 200, 203). We were pleased to find that the results obtained by multiblock chemometric analysis of six randomly prepared octonary chiral mixtures were in excellent agreement with the actual sample compositions.

Averaged $R^2$ and RMSE values for all set samples were determined as 0.96 and 0.26–0.28, respectively (Fig. 5b and Supplementary Tables 3–8). For example, the test set #1 contained 4.13 mM (R)-**PMP**, 1.38 mM (S)-**PMP**, 3.38 mM (R)-**PEA**, 1.13 mM (S)-**PEA**, 3.38 mM (1R, 2S)-**PPA**, 1.13 mM (1S, 2R)-**PPA**, 4.13 mM (R)-**PGL**, and 1.38 mM (S)-**PGL**. As shown in the Table in Fig. 5, MBPLS analysis of the coumarin sensing data predicted 4.06 mM (R)-**PMP**, 1.29 mM (S)-**PMP**, 3.30 mM (R)-**PEA**, 1.29 mM (S)-**PEA**, 3.30 mM (1R, 2S)-**PPA**, 1.33 mM (1S, 2R)-**PPA**, 4.07 mM (R)-**PGL**, and 1.30 mM (S)-**PGL**. Meanwhile, MBPCA + OLS predicted 3.98 mM (R)-**PMP**, 1.40 mM (S)-**PMP**, 3.62 mM (R)-**PEA**, 1.00 mM (S)-**PEA**, 3.62 mM (1R, 2S)-**PPA**, 1.00 mM (1S, 2R)-**PPA**, 3.98 mM (R)-**PGL**, 1.40 mM (S)-**PGL** (Supplementary Tables 16–19 and 46–49). As an alternative to multiblock analysis, we also investigated a combination of Least Absolute Shrinkage and Selection Operator (LASSO), which allows variable selection and traditional chemometric PCR and PLS regression. The benefit of this approach is that variable selection can improve the performance and interpretability of the model as irrelevant and noisy spectroscopic data are removed (Supplementary Tables 50, 117–126 and Supplementary Figs. 212–235)[36]. Although this proved quite successful, a closer

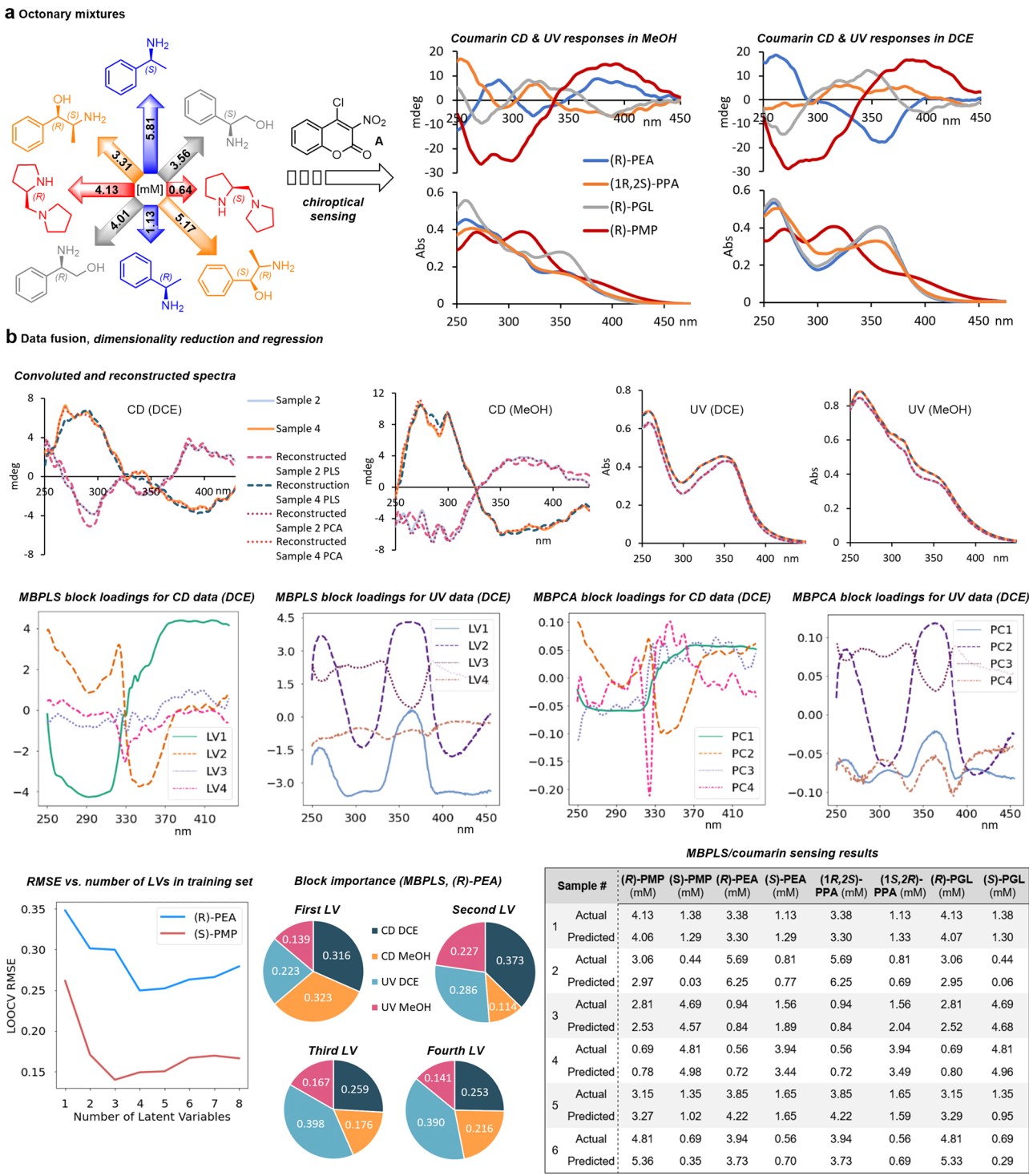

**Fig. 5 Multi-modal optical chirality sensing of octonary samples. a** Representation of a complex octonary chiral compound mixture and individual coumarin UV/CD responses for each chiral analyte in MeOH and DCE. **b** Top: The convoluted UV and CD spectra of five samples were subjected to PCA and PLS analysis and the reconstituted spectra were generated using inverse transformation. Results with two samples are shown. Middle: Representative MBPCA and MBPLS block loadings for CD and UV spectra in DCE for (*R*)-**PEA**. Bottom: Averaged LOOCV RMSE of the training sets vs. number of latent variables for (*R*)-**PEA** and (*S*)-**PMP**. The pie charts display the MBPLS block importance of 4 latent variables for (*R*)-**PEA**. The table shows the comparison of the octonary sensing results obtained with MBPLS for six test samples and the actual concentrations. **PEA**: 1-phenylethylamine, **PMP**: 1-(pyrrolidin-2-ylmethyl)pyrrolidine, **PGL**: phenylglycinol, **PPA**: 2-amino-1-phenylpropan-1-ol. MBPCA multiblock principal component analysis, MBPLS multiblock partial least square, DCE dichloroethane, LOOCV leave-one-out cross-validation, RMSE root mean squared error, LV latent variable. The data underlying this figure are available as Source Data.

inspection of the overall performance of all chemometric tools developed herein shows that the multiblock methods give superior results (Supplementary Tables 3–13, 51–58, 127, 128). Finally, we compared the effects of different scaling methods. We found that unit variance and multiblock scaling give results that generally have comparable error margins. The chemometric data obtained with unit variance scaling are shown in Fig. 5 and are discussed above. The results acquired with hard and soft block scaling using either LASSO + PCR or MBPLS for the regression analysis are provided in Supplementary Tables 59–116.

In summary, we have introduced organic reaction-based multimodal optical chirality sensing methodology and chemometric tools capable of orthogonal data fusion and spectral deconvolution to achieve stereochemical analysis of complicated mixtures of structurally analogous and therefore particularly challenging chiral target compounds. Reduction of the sensing data dimensionality coincides with minimal information loss as verified by accurate reconstruction of the original spectra thus setting the stage for efficient multiblock regression analysis. The practicality and speed of this approach were demonstrated with the determination of the absolute configuration, enantiomeric ratios, and individual concentrations of quaternary and octonary samples with drastically varying chemical and chiral compositions. The successful development of straightforward chemometric in situ chirality sensing methodology using a simple achiral probe and an optimized MBPLS algorithm described herein overcomes major obstacles originating from the difficulty with quantitative deconvolution of a massive amount of spectroscopic data that are generated by simultaneous detection of several chiral analytes and the low resolution of inherently broad and largely overlapping CD and UV absorption bands, a complexity that has not been solved previously. In the future, it seems likely that the integration of chiroptical sensing and chemometric technologies will supersede traditional chirality analysis workflows and drastically accelerate the discovery pace in numerous academic and industrial settings.

## Data availability

The data generated in this study are provided in the Supplementary Information/Source Data file. Source data are provided with this paper.

## Code availability

Data handling and processing were conducted in Python using pandas (https://github.com/pandas-dev/pandas), numpy (https://github.com/numpy/numpy), and scikit-learn (https://github.com/scikit-learn/scikit-learn) except for block scaling which was run in R using the software package prospectr available at https://cran.r-project.org/web/packages/prospectr/index.html. Linear regression, LASSO, PCA, and PLS were conducted in Python using scikit-learn available at https://github.com/scikit-learn/scikit-learn. MBPCA was performed in R using the software package Multiblock Sparse Multivariable Analysis (msma) available at https://CRAN.R-project.org/package=msma. MBPLS was performed in Python using the software package Multiblock Partial Least Squares available on github at https://github.com/DTUComputeStatisticsAndDataAnalysis/MBPLS.

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

## Acknowledgements
We gratefully acknowledge financial support from the U.S. National Science Foundation (CHE-1764135, C.W.).

## Author contributions
D.S.H. and C.W. designed and conceived the experiments. D.S.H performed the experiments and the chemometric analyses. D.S.H. and C.W. discussed the results and wrote the paper.

## Competing interests
The authors declare no competing interests.
