## [Peer Review File · Nature Communications]

REVIEWER COMMENTS

Reviewer #1 (Remarks to the Author):

The manuscript addresses the possibility of coupling chemometric data processing with UV and CD spectroscopy for the quantification of enantiomers in mixtures. The authors critically discuss the state of the art and thoroughly address the limitation of standard univariate approaches, and the advantages given by multivariate and, eventually, multi-block approaches. Overall, the experimental part is extensively discussed and the results of data analysis are clearly presented and appear to be rather accurate.

For all these reasons I do think that the manuscript deserves publication on the journal. Prior to final acceptance, however, I would suggest the authors to address the following minor issues:

- 1) Page 9, 14 lines from the bottom: "random but correlated" can be a misleading statement, since the main characteristic of randomly varying variables is to be not correlated to one another.
- 2) Why PLS was not used as "single-block" modeling technique (see below also for a comment on this), whereas it was considered in the multi-block configuration?
- 3) When using OLS on the concatenated PCs from the UV and CD blocks, this is already a multi-block modeling; in particular, it is an example of mid-level data fusion. In this context, I was wondering whether the number of principal components to be selected from each of the block was optimized in any way, e.g., by cross-validation, and how did the authors cope with the possibility that one of the blocks could dominate just by the fact of carrying a higher amount of variance. Was there any block-scaling performed? The same questions holds for the multi-block PCA and PLS part. Scaling is a fundamental issue in multi-block analysis and it should be commented.
- 3) Was the lasso regression implemented on the concatenated UV and CD matrices? How was the optimal value of the lambda parameter selected? What was its optimal value? How many variables were showing regression coefficients significantly different from zero?
- 4) Have the authors considered adopting other possible variable selection strategies such as the ones based on VIP scores/selectivity ratio or specifically designed for multi-block data as SO-CovSel?

Reviewer #2 (Remarks to the Author):

By merging chiroptical detection methodology and artificial intelligence for spectral deconvolution, Hassan and Wolf report a protocol that allows the complete analysis (i.e. absolute configuration, enantiomeric ratios, and individual concentrations) of complex mixtures of chiral amines. The approach is original and proved, in this study, to be very successful. Such an approach could definitely become a standard in the future thanks to a couple of advantages regarding practicality, speed, and accuracy.

As usual with this research group, the paper is well written, the data are solid, and the conclusions are convincing. Accordingly, I recommend publishing this paper as it.

One minor comment: the authors state that high-throughput screening applications with margins of error of 10% are considered satisfactory. I wonder where this number comes from, and if the authors can provide more details to back up this claim.

Reviewer #3 (Remarks to the Author):

1. I personally think the abstract is too long and can be reduced, for example, you can just explain what is presented in this work, the main experiment etc. currently there is lot of background in abstract

2. There are no-line numbers in the manuscript so I cannot do a detailed review, please put them in future submissions

3. In the introduction in almost all major places, I noted that citation are used collectively, for example, a statement is made and several citation are accumulated, I will recommend to critically cite the articles individually to demonstrate the contribution of each cited article.

4. I did not understand this : "One simple yet powerful LVMR method is Principal Component Regression (PCR), which is a

multivariate calibration technique that combines PCA with Ordinary Least Squares (OLS) " did you not use the PLS regression here?

5. I do not understand this : "We then used two multi-block algorithms, MBPCA and MBPLS, for regression analysis". MBPCA is just a decomposition method probably you must have combined it with OLS , on other hand MBPCA and PLS are highly dependent on the way you have pre-processed the data blocks, I do not find much information about the specific pre-processing?

6. Furthermore, for regression task you should use So-PLS type of analysis which does not depend on the scales and only models complementary information data sets being fused

7. I do not understand. "In summary, we have introduced organic reaction based multi-modal optical chirality sensing

methodology and machine learning " what is machine learning here? you are using chemometric tools here, for example, multi-block is developed in the chemometric community, Please replace such misnomers all along the text

8. in conclusions "an optimized MBPLS algorithm" : how did you optimised the MBPLS, I did not find much information,

I think overall the article is well written and all the graphics are well presented, but, please clarify above points before its publication.

Responses to the reviewers:

Reviewer #1 (Remarks to the Author):

The manuscript addresses the possibility of coupling chemometric data processing with UV and CD spectroscopy for the quantification of enantiomers in mixtures. The authors critically discuss the state of the art and thoroughly address the limitation of standard univariate approaches, and the advantages given by multivariate and, eventually, multi-block approaches. Overall, the experimental part is extensively discussed and the results of data analysis are clearly presented and appear to be rather accurate.

For all these reasons I do think that the manuscript deserves publication on the journal. Prior to final acceptance, however, I would suggest the authors to address the following minor issues:

1) Page 9, 14 lines from the bottom: "random but correlated" can be a misleading statement, since the main characteristic of randomly varying variables is to be not correlated to one another.

We have changed this statement accordingly to "Figure 2 contains an illustration of how PCA can transform a 2D plot with correlated data points into one dimension".

2) Why PLS was not used as "single-block" modeling technique (see below also for a comment on this), whereas it was considered in the multi-block configuration?

We were already satisfied with the performance of PCR and the quaternary mixture sensing analysis was an initial test to see if a chemometric approach with a coumarin sensor would work. As we show in Figure 2 and discuss in the text, the results obtained with our PCR coumarin sensing analysis are very close to the original sample compositions. With this success in hand, we decided to move on to the octonary mixture challenge.

3) When using OLS on the concatenated PCs from the UV and CD blocks, this is already a multi-block modeling; in particular, it is an example of mid-level data fusion. In this context, I was wondering whether the number of principal components to be selected from each of the block was optimized in any way, e.g., by cross-validation, and how did the authors cope with the possibility that one of the blocks could dominate just by the fact of carrying a higher amount of variance. Was there any block-scaling performed? The same questions holds for the multi-block PCA and PLS part. Scaling is a fundamental issue in multi-block analysis and it should be commented.

We actually used cross-validation in the quaternary and octonary mixture analysis. This is mentioned on pages 9-10 " Cross-validation was performed on the training sets to evaluate how well the model can be applied to new data (SI). " and on page 12: "In order to maximize accuracy and minimize overfitting at the same time, the optimal number of latent variables/principal components having the smallest RMSE was determined by leave-one-out cross-validation (LOOCV)." See also S148-161.

We agree with the reviewer that a deeper analysis of the effects of different scaling methods is important. We now compare unit variance scaling which we used originally with multiblock scaling results. The data from both hard and soft block scaling using either LASSO + PCR or MBPLS for regression (lambda and number of latent variables are optimized through cross-validation) are now included in the SI, see S170-221. Overall, the

scaling methods result in comparable performances. We therefore added the following statement in the manuscript on page 12: “Finally, we compared the effects of different scaling methods. We found that unit variance and multiblock scaling give results that generally have comparable error margins. The chemometric data obtained with unit variance scaling are shown in Figure 3 and are discussed above. The results acquired with hard and soft block scaling using either LASSO+PCR or MBPLS for the regression analysis are provided in the SI.”

3) Was the lasso regression implemented on the concatenated UV and CD matrices? How was the optimal value of the lambda parameter selected? What was its optimal value? How many variables were showing regression coefficients significantly different from zero?

LASSO was originally implemented on the concatenated matrices and we were using the default lambda value (0.01) without further optimization. We have now optimized lambda values using 5-fold cross-validation individually for each analyte. The optimized values, number of variables with regression coefficients significantly different from zero, and the results obtained with LASSO+PCR and LASSO+PLS are shown in the SI, see S211-221. We now have the following statement in the discussion on page 12: “As an alternative to multiblock analysis, we also investigated a combination of Least Absolute Shrinkage and Selection Operator (LASSO) which allows variable selection and traditional chemometric PCR and PLS regression. The benefit of this approach is that variable selection can improve performance and interpretability of the model as irrelevant and noisy spectroscopic data are removed (see SI for details including optimization of lambda values for each analyte). Although this proved quite successful, a closer inspection of the overall performance of all chemometric tools developed herein shows that the multiblock methods give superior results.”

4) Have the authors considered adopting other possible variable selection strategies such as the ones based on VIP scores/selectivity ratio or specifically designed for multi-block data as SO-CovSel?

In this study we investigated MBPCA, MBPLS and a combination of LASSO with PCR or PLS regression. We were pleased to find that these give very good results with the challenging octonary mixtures. We understand that other methods can also be very useful and we consider using SO-CovSel in future applications.

Reviewer #2 (Remarks to the Author):

By merging chiroptical detection methodology and artificial intelligence for spectral deconvolution, Hassan and Wolf report a protocol that allows the complete analysis (i.e. absolute configuration, enantiomeric ratios, and individual concentrations) of complex mixtures of chiral amines. The approach is original and proved, in this study, to be very successful. Such an approach could definitely become a standard in the future thanks to a couple of advantages regarding practicality, speed, and accuracy.

As usual with this research group, the paper is well written, the data are solid, and the conclusions are convincing. Accordingly, I recommend publishing this paper as it.

One minor comment: the authors state that high-throughput screening applications with margins of error of 10% are considered satisfactory. I wonder where this number comes from, and if the authors can provide more details to back up this claim.

This error margin was developed through several discussions with industrial chemists interested in high-throughput screening of asymmetric reactions. We now cite reference 9 and add reference 32 to back up the 5-10% range.

Reviewer #3 (Remarks to the Author):

1. I personally think the abstract is too long and can be reduced, for example, you can just explain what is presented in this work, the main experiment etc. currently there is lot of background in abstract

We agree and have reduced the abstract to ~150 words which is also a journal requirement.

2. There are no-line numbers in the manuscript so I cannot do a detailed review, please put them in future submissions

We understand that this is a problem and apologize. However, we have experienced that the resolution of the article is severely compromised when we use the online pdf conversion tool to add line numbering.

3. In the introduction in almost all major places, I noted that citation are used collectively, for example, a statement is made and several citation are accumulated, I will recommend to critically cite the articles individually to demonstrate the contribution of each cited article.

We sometimes cite a few references together which in our opinion gives due recognition to these carefully selected papers without unnecessarily extending the introduction. We believe that individual discussions of each contribution would be beyond the scope of this article.

4. I did not understand this : "One simple yet powerful LVMR method is Principal Component Regression (PCR), which is a multivariate calibration technique that combines PCA with Ordinary Least Squares (OLS) " did you not use the PLS regression here?

We did not apply PLS regression in the quaternary mixture analysis because we were already satisfied with the performance of PCR.

5. I do not understand this : "We then used two multi-block algorithms, MBPCA and MBPLS, for regression analysis". MBPCA is just a decomposition method probably you mist have combined it with OLS , on other hand MBPCA and PLS are highly dependent on the way you have pre-processed the data blocks, I do not find much information about the specific pre-processing?

We now provide more information on the pre-processing (unit variance scaling vs block scaling). This is mentioned in the new text on page 12. See also S186-221.

6. Furthermore, for regression task you should use So-PLS type of analysis which does not dependent on the scales and only models complimentary information data sets being fused

We understand that many chemometric methods can in principle be used in conjunction with the reaction based sensing methodology described in our study. We decided to use MBPCA, MBPLS, LASSO+ PCR and LASSO+PLS for this task and obtained very good results. We notice, however, the value of SO-PLS analysis which is invariant to the relative scaling of the input blocks. Similarly, we found that regardless of the scaling (block or unit variance scaling), the octonary mixture sensing results seem to be comparable when we use MBPLS.

7. I do not understand. "In summary, we have introduced organic reaction based multi-modal optical chirality sensing methodology and machine learning " what is machine learning here? you are using chemometric tools here, for example, multi-block is developed in the chemometric community, Please replace such misnomers all along the text

As suggested, we replaced the term "machine learning" with "chemometrics" everywhere in the text and figures of the manuscript.

8. in conclusions "an optimized MBPLS algorithm" : how did you optimised the MBPLS, I did not found much information,

This is discussed on page 12: "In order to maximize accuracy and minimize overfitting at the same time, the optimal number of latent variables/principal components having the smallest RMSE was determined by leave-one-out cross-validation (LOOCV). For example, three and four LVs were used for the analysis of (S)-PMP and (R)-PEA, respectively (Fig. 3b and SI)." See also S148-162.

I think overall the articles is well written and all the graphics are well presented, but, please clarify above points before its publication.

REVIEWERS' COMMENTS

Reviewer #1 (Remarks to the Author):

I think that the authors replied satisfactory to my own comments and to the ones of the other reviewers and therefore the manuscript can now be considered suitable for acceptance